



# First measurements of 3-Dimensional winds up to 25 km based on Aerosol backscatter using a compact Doppler lidar with multiple fields of view

Thorben H. Mense[1], Josef Höffner[1], Gerd Baumgarten[1], Ronald Eixmann[1], Jan Froh[1], Alsu Mauer[1], Alexander Munk[2], Robin Wing[1], and Franz-Josef Lübken[1]

[1]Leibniz Institute of Atmospheric Physics at the University of Rostock, Kühlungsborn, Germany, 18225
[2]Fraunhofer Institute for Laser Technology ILT, Aachen, Germany, 52074

**Correspondence:** Thorben H. Mense (mense@iap-kborn.de)

**Abstract.** We present the first measurements of simultaneous horizontal and vertical winds using a new VAHCOLI lidar system developed at the Leibniz Institute for Atmospheric Physics in Kühlungsborn, Germany (54.12°N, 11.77°E). We describe the technical details of a multi-field-of-view (MFOV) upgrade, which allows to measure wind dynamics in the transition region from microscale to mesoscale ($10^3 - 10^4$ m). The method was applied at the edge of a developing high-pressure region.

Comparisons between the lidar measurements and data from ECMWF show excellent agreement for the horizontal wind components, better than $0.30 \pm 0.36$ m s$^{-1}$ along the north beam of the lidar and $-0.93 \pm 0.73$ m s$^{-1}$ along the south beam. Measurements of vertical wind show significant underestimation of this component by ECMWF. Comparison to ADM-Aeolus shows good agreement, better than $-0.12 \pm 3.31$ m s$^{-1}$. The capability of the MFOV lidar to explore small-scale asymmetries in the wind field is shown by comparison of the north and south field of view, where we observe a wind asymmetry in the

meridional winds, which is also present in ECMWF but underestimated by a factor of approximately four.

## 1 Introduction

Accurate and comprehensive measurements of wind profiles play a critical role in various atmospheric studies and applications, including numerical weather prediction (NWP) (Baker et al., 2014; Stoffelen et al., 2005). However, obtaining such measurements presents significant challenges. In situ measurements using anemometers mounted on towers or masts are limited to the

boundary layer, whereas wind measurements from radiosondes, aeroplanes, and rockets can only offer snapshots of the winds, lacking continuous coverage.

To achieve continuous wind measurements with a high degree of accuracy and spatiotemporal resolution, ground-based remote sensing methods are required. The most prominent ground-based techniques for measuring wind are sodar, radar, and lidar. Sodar and radar face limitations in their altitude coverage due to technical constraints. Sodar wind profilers reach heights

of up to two kilometres by scattering sound waves off turbulence (Bailey, 2000), while radar can measure the troposphere and in the upper mesosphere and lower thermosphere (MLT) due to the presence of available scattering targets. However, there exists a significant altitude gap in radar's capability to measure wind, between 20 and 80 km (Hocking, 1997). Ground-based



Doppler-wind lidars (DWL) are the only remote sensing instruments capable of providing continuous wind profiles from the troposphere to the MLT with high vertical and temporal resolution.

One challenge associated with Doppler-wind lidars is their inherent complexity, necessitating ample laboratory space for emitters, detectors, and telescopes, as well as skilled operators. Presently, there are only six operational DWLs worldwide capable of measuring wind in the middle atmosphere. These installations are located at L'Observatoire de Haute Provence (OHP) in France (44°N, 6°E) (Souprayen et al., 1999b, a), Observatoire de Physique de l'Atmosphère de la Réunion (OPAR) on La Reunion island (21°S, 55°E) (Ratynski et al., 2022), the Arctic Lidar Observatory for Middle Atmosphere Research

(ALOMAR) in Norway (69°N, 16°E) (Baumgarten, 2010), the Leibniz Institute for Atmospheric Physics in Kühlungsborn, Germany (54°N, 12°E) (Wing et al., 2022), and two Mobile Wind lidars in the Hefei region of China (approximately 32°N, 117°E) (Xia et al., 2012; Yan et al., 2017; Chen et al., 2023).

Space-based Doppler wind lidars provide the ability to obtain global wind profiles. The first satellite carrying such an instrument is Atmospheric Dynamics Mission-Aeolus (ADM-Aeolus), which embarked on its mission in August 2018 and

continued its measurements until May 2023. Aeolus orbits the Earth in a Sun-synchronous, dusk/dawn orbit (inclination of 97°) at a height of 320 km (ESA, 2020). Winds are measured orthogonal to the flight direction at an angle of 35 degrees off-nadir on the night side of the Earth. Along this field of view, the mission aimed to provide line of sight (LOS) winds from the ground up to the lower stratosphere (up to 30 km) with an altitude resolution of 250 m to 2 km and an altitude-dependent precision of $1\,\mathrm{m\,s^{-1}}$ to $3\,\mathrm{m\,s^{-1}}$ (Drinkwater et al., 2016). The actual error estimates during the operation were proved to be in

the range of $4.1\,\mathrm{m\,s^{-1}}$ to $4.4\,\mathrm{m\,s^{-1}}$ (Rayleigh) and $1.9\,\mathrm{m\,s^{-1}}$ to $3.0\,\mathrm{m\,s^{-1}}$ (Mie) by Martin et al. (2021).

The Atmospheric Laser Doppler Instrument (ALADIN) onboard Aeolus used a frequency tripled Nd:YAG laser operating at 355 nm and two distinct receiver channels. The first channel utilises coupled Fabry-Pérot interferometers to capture the Rayleigh scattering signal, while the second channel incorporates a Fizeau interferometer to detect the narrowband Mie scattering (Paffrath et al., 2009; Reitebuch et al., 2009).

Since the wind measurements presented in this work relies solely on Mie-scattering to cover the UTLS a closer look on its use by the Aeolus satellite is valuable. Within the height range covered by Aeolus, the Rayleigh and Mie scattering techniques serve complementary purposes. In scenarios where the atmosphere is laden with significant aerosol loads, such as clouds or smoke plumes, the quality of the Rayleigh winds may be compromised. However, the Mie winds derived under these conditions exhibit exceptional quality, with error estimates approximately half of that associated with clean Rayleigh winds (Rennie et al.,

2021; Rani et al., 2022).

This reduction in measurement error can be attributed to the narrowband nature of Mie scattering, which allows for a more accurate estimation of the Doppler shift. Due to the substantial mass of the aerosol scattering particles, temperature-induced broadening of the Mie backscattered signal can be neglected. As a result, the spectral width of the Mie scatter is about two orders of magnitude smaller than that of the Rayleigh scatter for Aeolus and about three orders of magnitude smaller for the

instrument presented in this work.

While these advantageous spectral properties of Mie scattering enable more precise wind measurements, it is important to note that the ALADIN instrument requires complementary Rayleigh-winds, since the use of the Mie channel necessitates





sufficient aerosol loading to derive Mie winds. Consequently, it cannot solely rely on background aerosols present in the atmosphere for wind measurements (Drinkwater et al., 2016; Reitebuch et al., 2009). However, as will be shown in the following

sections, the lidar systems introduced in this work have the unique capacity of detecting aerosols that are invisible to ALADIN, which makes a cluster of them especially interesting for intercomparison studies to upcoming Aeolus successors. Additionally, such a cluster or network will be especially interesting to observe atmospheric phenomena on horizontal and vertical scales which are inaccessible to other measurement techniques, as discussed in Lübken and Höffner (2021).

    The organisation of this paper is as follows: Section 2 provides a technical overview of the lidar system, including its de-

sign, instrumentation, and key features. Section 3 describes a measurement campaign conducted from the 16th to the 18th of December 2022, data collection procedures, and processing techniques. Section 4 presents the results of the wind measurements, including the vertical and horizontal wind profiles obtained and their comparison to data from the European Centre for Medium-Range Weather Forecasts (ECMWF) and Aeolus. Finally, Section 5 discusses the implications of the findings, highlights the contributions of this work, and suggests potential avenues for future research.

**2    The lidar system**

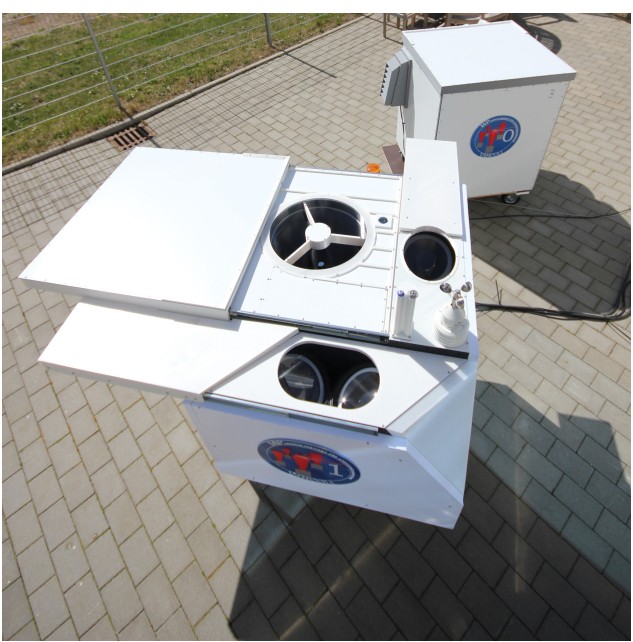

**Figure 1.** The upgraded VAHCOLI 1 lidar system on the testing ground in Kühlungsborn, Germany. The automatic hatches are open, revealing the 50 cm vertical pointing mirror telescope visible in the centre of the core unit and the four additional lens telescopes mounted to two sides of the core unit. Other interesting features visible include the window for the vertical pointing camera, next to the vertical telescope and the weather station in the corner of the MFOV upgrade. The prototype lidar unit (VAHCOLI 0), which can only measure vertically, is visible in the background (Lübken and Höffner, 2021).



The lidar unit used in this work is called "VAHCOLI 1". Figure 1 shows this instrument, along with a prototype system. VAHCOLI stands for Vertical And Horizontal Coverage by LIdar and is a concept for lidars which encompasses a network of instruments to provide the horizontal coverage lidars typically lack (Lübken and Höffner, 2021). For this, a new lidar system has been developed, which is small and mobile, yet designed to measure Doppler-Mie, Doppler-Rayleigh and Doppler-resonance

scatter. The first prototype of this new lidar system (VAHCOLI 0) started operation in late 2019 and since then, two additional lidar systems have been constructed (VAHCOLI 1 and 2). In 2022, the capabilities of VAHCOLI 1 was significantly extended with a multi-field-of-view (MFOV) upgrade to provide a new way to measure horizontal and vertical winds simultaneously. Though the general measurement principle of the VAHCOLI lidar has been developed based on the mobile potassium lidar of IAP the measurement hardware has drastically improved in order to have such a compact system (von Zahn and Höffner, 1996;

Fricke-Begemann et al., 2002; Höffner and Lübken, 2007).

## 2.1 The lidar core unit

The core unit of a VAHCOLI 1 comprises all the hardware necessary for fully autonomous measurements in a rugged, climate-controlled enclosure with a volume of roughly one cubic meter. Figure 2 illustrates how this enclosure is built in modular layers that are 3D printed in-house using large-scale FFF (fused filament fabrication) printers. Likewise, the majority of the

internal structural components are printed in-house. Layer 1 contains housekeeping hardware, power supplies for the system, an battery to buffer short power grid issues and a custom industrial computer, which controls the whole system with three high-performance measurement cards. Layer 2 contains parts of the optical setup, like the seeding laser, the diode pump and a fibre resonator for reference, as well as the compact detection bench with its two avalanche photodiode (APD) detectors and stabilised filters. Layer 3 houses the emission part, with the biggest part being the diode-pumped Alexandrite ring laser, which

was specially designed in close collaboration with the Fraunhofer Institute for Laser Technology in Aachen (Munk et al., 2021). The telescope cylinder, for the vertically facing 50 cm mirror telescope, passes through all layers. A more detailed description of the core unit and the measurement technique can be found in (Lübken and Höffner, 2021) and (Froh, 2021).

## 2.2 The multi-field-of-view upgrade

In order to expand the capabilities of the core unit to measurements of horizontal winds, the system was upgraded with multiple

fields of view ("MFOV upgrade"). While other Doppler-lidars that measure at stratospheric altitudes use one telescope for each wind component, VAHCOLI 1 has four tilted telescopes that allow it to measure each horizontal component with two opposing fields of view (Khaykin et al., 2020; Baumgarten, 2010; Wing et al., 2022). This allows the direct observation of gradients in the measured atmospheric parameters, by the comparison of two fields of view.

The general optical setup of the MFOV upgrade can be divided into a 2-axes galvanometer scanner (inside the core unit) and

an L-shaped enclosure holding the four additional telescopes, which is mounted to the walls of the core unit. In the core unit of VAHCOLI 1, the galvanometer scanner enables rapid switching between the five fields of view in less than 1 ms, by rapidly rotating its two mirrors to pre-programmed positions.

The following specifications were met in the design of the four additional telescopes: (1) Compact and relatively light design



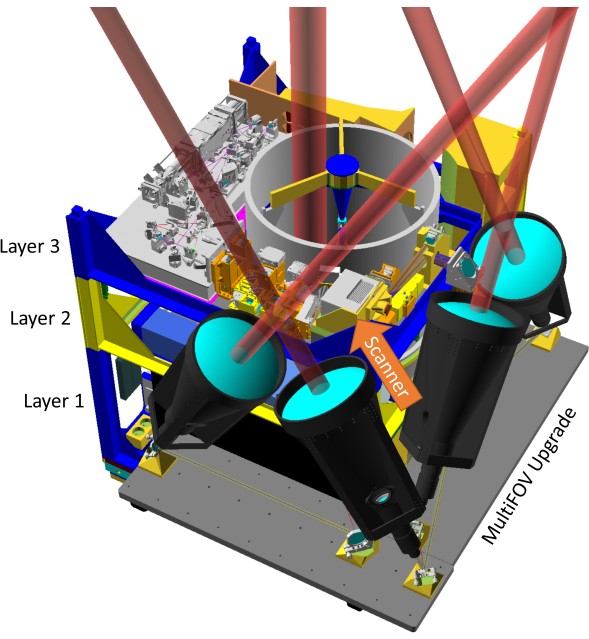

**Figure 2.** CAD drawing of the new lidar unit equipped with multiple fields of view. A cubic meter core unit contains the measurement hardware of the lidar and a 50 cm mirror telescope looking vertically similar to the system described in Lübken and Höffner (2021). The optical setup in the core unit has been extended by a two-axes galvanometer scanner, controlling the field of view. Four additional telescopes are mounted in an L-shaped extension mounted to the side of the system. The outgoing laser beams of the system are represented in red and the three individual layers of the core unit are colour coded in blue and yellow.

to keep the dimensions of the upgrade small and forces on the core unit low. (2) Reasonable large aperture to measure up to

the stratosphere. (3) A cost-effective solution to allow for network construction as envisaged in VAHCOLI. Since no available telescopes on the market met these requirements, we developed them to the specifications we needed. We designed diffraction-limited lens telescopes with a clear aperture of the front lens of 200 mm, an overall length smaller than 600 mm and a weight of less than 5 kg. We use commercial off-the-shelf and 3D-printed components, allowing for fast and cost-effective integration of the MFOV upgrade. A challenge we faced in the telescope design was caused by the requirement to operate the system

night and day. This presented the risk of the telescopes pointing directly at the sun, resulting in considerable thermal load. To overcome this obstacle, we implemented a robust cooling system and carefully selected materials that ensures the telescopes' resilience in such circumstances. Four of these telescopes are held in the 3D-printed enclosure of the MFOV upgrade at a $30°$ off zenith angle, facing north, south, east and west. The pointing precision of the telescopes is limited only by the precision of the 3D printer since the enclosure and telescope mount are printed in one big part. A tilt sensor inside the core unit measures

the level of the whole unit with a precision better than $0.1°$, and an integrated camera can be used to additionally verify the instrument orientation using the stars with a precision better than $0.2°$.





The MFOV upgrade drastically increases the capabilities, while the hardware upgrades were manageable. The high-speed 2-axes scanner and the four extra fields-of-view, however, required significant upgrades to the control software.

## 3 The December 2022 campaign

The first campaign using VAHCOLI 1 with the MFOV upgrade took place in December 2022, namely from the 16th of December at 11:28 UT until the 18th of December at 12:10 UT.

### 3.1 Measurement conditions

To acquire the measurements presented in this publication, the VAHCOLI 1 instrument was placed outside on a paved and fenced-in area, with markings on the ground indicating true north direction. No further preparations of the site were required.
The orientation of the system in the true north direction was verified by targeting Polaris. During the measurement, the local weather conditions were monitored by a weather station situated roughly 50 m away from the instrument. Over the course of the measurement, the air pressure increased from 1000 hPa to 1024 hPa, the ground temperature was between -2 °C and -8 °C and the relative humidity was high with values ranging between 75 and 100%.

At the beginning of the measurement, the signal strength of all four oblique fields of view was about a factor of 3 to 4 less
than that of the vertical field of view (see Figure 3). This is expected due to the smaller size of the lens telescopes. Over the course of the measurement, the signal in the meridional fields of view stayed almost constant, whereas the zonal fields of view experienced a sharp loss in signal strength. The signal decreased due to icing and condensation occurring on the two zonal pointing telescopes, caused by the cold temperatures and high relative humidity. An internal heating element was switched on for de-icing around 12:00 UT on the 17th, leading to an increase in the signal of both telescopes. The signal in the
westward pointing telescope only recovered to 13 % of its initial value. As a result, our westward zonal winds have a lower top altitude than our eastward zonal winds and both have a significantly reduced performance with more data gaps compared to the meridional telescopes.

Except for the icing problem, the system and the MFOV upgrade performed well with no major problems or interruptions, even under these harsh winter conditions close to the Baltic Sea. After this initial measurement campaign, the VAHCOLI 1 system
remained in operation and, as of June 2023, has accumulated more than 600 hours of measurements.

### 3.2 Retrieval of horizontal and vertical winds from the lidar data

VAHCOLI is a frequency scanning lidar, which measures the spectra of the backscattered signal (Höffner et al., 2021; Lübken and Höffner, 2021). For the measurements presented in this publication, the Mie channel of the system has been used. Mie scattering provides the best wind measurements up to 25 km due to its very narrow spectrum, but is limited in the lower
stratosphere by the decreasing concentration of aerosols. Above typically 25 km, the aerosol density drastically drops, giving an upper altitude limit for the use of this method.

For data processing the individual pulse raw data, produced by the system with a time resolution of 2 ms and an altitude





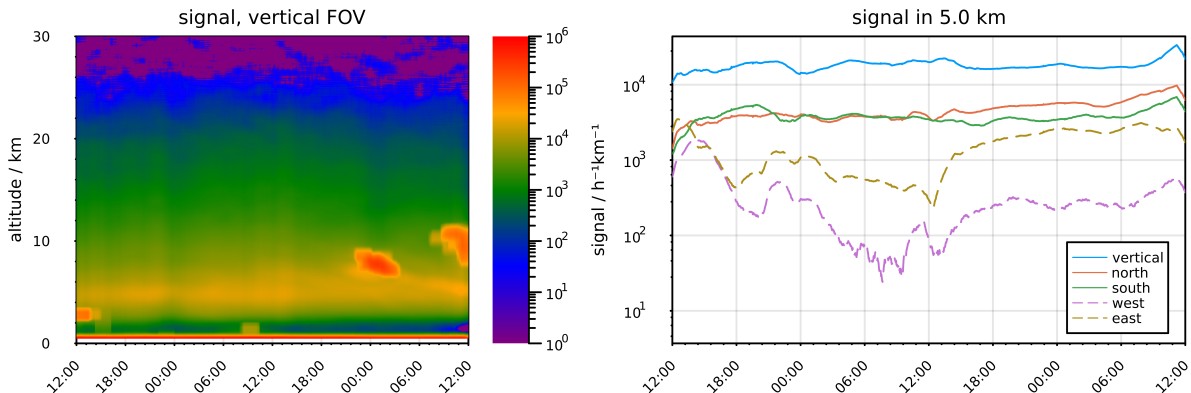

**Figure 3.** Backscatter signal of the vertical field of view (left) and backscatter signal at 5 km in each of the five fields of view (right) from 16 December 2022 12:00 UT to 18 December 2022 12:00 UT. The data is smoothed with 2 hours and 1 km running mean filter and given in counts per time ($\Delta t = 2$ hours) per altitude ($\Delta z = 1$ km). The signal increase at approximately 7-11 km at 00:00 on December 18th is caused by the presence of cirrus clouds, which effectively improves the capability to measure winds with the lidar. Due to a larger aperture the signal from the vertical FOV is generally larger compared to the horizontal FOVs.

resolution of 1 m, are first converted to backscatter spectra with an integration time of 2 minutes and a height resolution of 1 m for each individual field of view. In order to improve the statistics for the analysis, these spectra are then integrated in time and

altitude with a floating window to the wanted final resolutions of the data. Unless stated otherwise, an integration window of 2 hours and 1000 m is used, which is moved in 2 minute and 100 m steps. Winds are derived from the spectra by fitting a Voigt-function to each height channel, with an example of this fit shown in Figure 5. The laser's spectral shape was determined to be a Voigt function through lab measurements, which will be detailed in an upcoming publication. The detection filter employed for Mie scattering is a confocal etalon, which has a Lorentzian filter curve. The atmospheric broadening, corresponding to wind

fluctuations, is assumed to follow a Gaussian distribution. Hence, the convolution of a Voigt function, a Lorentzian function, and a Gaussian function yields another Voigt function.

From the Doppler-shift, $\Delta f$, given by fitting a Voigt function to the measured spectra, the line of sight wind can be derived by rearranging the Doppler-shift equation to $v_{LOS} = \Delta f/(2f_0) \cdot c$ with $c$ the speed of light and $f_0$ the fundamental frequency (e.g. the laser frequency). Since we measure both the vertical and four off-zenith line of sight winds simultaneously, the vertical

component can be immediately removed to derive horizontal wind components. The horizontal winds must be divided by $\sin(30°)$ to compensate for the off-zenith tilt. It is important to note, that this technique is based on a relative measurement of the Doppler-shift. This is done by comparing the measured backscatter spectrum to the reference spectrum within the system, as described by Lübken and Höffner (2021). For Mie-Doppler measurements no absolute frequency measurement is required and the fundamental wavelength of the laser is known to be 769.898 nm by the laser settings and the interference filters in the

system.

The error estimate for the line of sight winds ($\sigma_{\Delta v}$) is derived from the error estimate of the Doppler shift ($\sigma_{\Delta f}$) using the





propagation of uncertainties ($\sigma_{\Delta v} = \sigma_{\Delta f} \cdot c/(2f_0)$). $\sigma_{\Delta f}$ is a product of the fit routine, which uses the least square fitting of the spectrum, with the standard deviation of each fit parameter being the square root of the corresponding diagonal element of the covariance matrix (Hansen et al., 2013). This requires an estimate for the error of the photon counts, ($n$), in each frequency
bin of the spectrum. For this, we assume a Poisson distribution with a width of $\sigma_n = \sqrt{n}$. The precision of the result of the fit benefits from the very narrow spectral shape of the received Mie signal and its high edge steepness which facilitates precise determination of the Doppler shift. Together with the suppression of the background Rayleigh signal, the number of needed photons is significantly reduced to achieve acceptable wind errors compared to traditional DWL systems (Höffner et al., 2021). Figure 4 shows this connection between the measured spectrum of the Mie scattering, the calculated line of sight wind and

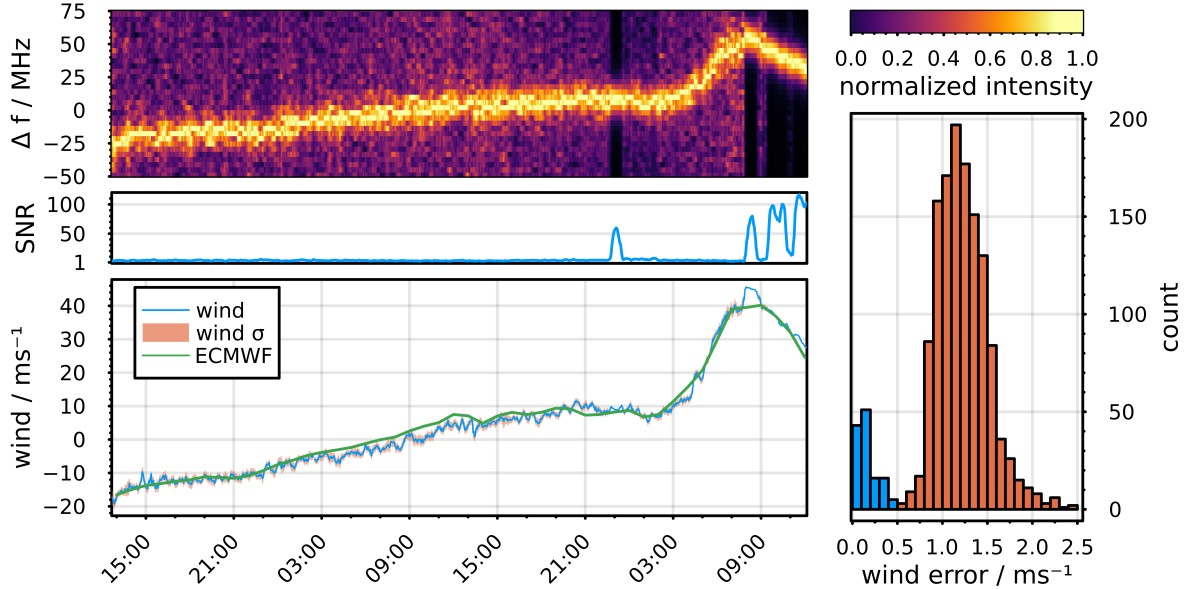

**Figure 4.** Connection between measured spectra, calculated wind and wind error. **top left:** Example of an atmospheric spectrum measured between the 16th and 18th of December 2022 with the north-facing telescope integrated from 9.25 km to 10.25 km, a gridded time resolution of 2 minutes and an integration time of 20 minutes. The purple and black regions in the spectrum plot represent the lidar background. The black corresponds to the cirrus clouds seen in Fig. 3 which result in a higher SNR value. **centre left:** Signal-to-noise ratio (SNR) calculated from the spectrum above. **top left:** Calculated lidar average wind speed (blue) together with the wind error estimate (orange ribbon), which most of the time at line strength of the wind speed. The corresponding ECMWF-IFS wind time series is shown for reference (green). A significant deviation of the measured wind and the ECMWF wind is visible at the 18th December around 8:00 UT, probably caused by the much higher smoothing in ECMWF. **bottom right:** Histogram of the error estimate for the wind from measurements with high (blue) and low (orange) aerosol loading.

the error estimate. A change of the peak position in the measured spectrum results in a change in the wind. Black lines in the background of the spectrum correspond to the presence of cirrus clouds, which significantly increase the signal to noise ratio of the spectrum and thus reduce the calculated error of the measurement. This leads to a double distribution of the wind error





visible in the bottom right of Figure 4. With one peak around 0.15 m s$^{-1}$ produced by measurements with cirrus clouds and one peak around 1.15 m s$^{-1}$ from measurements without cirrus clouds. Figure 5 shows an examples of each of these cases.

The clear air case shows a low signal to noise ratio of 4.2, resulting in an error estimate of the Doppler-shift of 1.6 MHz and thus a wind error of 0.6 m s$^{-1}$. In the case of a cirrus cloud the signal to noise ratio drastically increases by more than an order of magnitude, resulting in an error estimate of the Doppler-shift of 0.2 MHz and thus a wind error of less than 0.1 m s$^{-1}$. In cirrus clouds and other strong aerosol loads the integration in time and altitude can be significantly reduced, while maintaining wind errors below 1 m s$^{-1}$.

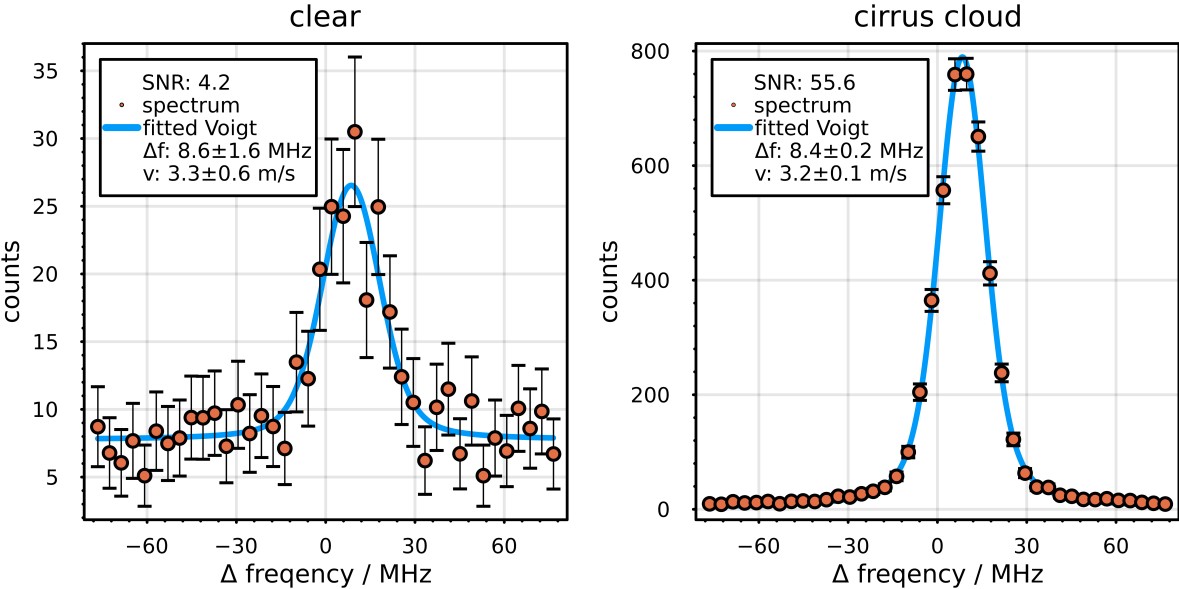

**Figure 5.** Example of the spectrum measured from 9.25 km to 10.25 km in the north field of view, together with the fitted Voigt function in clear air (left) and in a cirrus cloud (right). Corresponding signal-to-noise ratios and errors in frequency offset and winds derived from the fits are shown in the inlets.

**4 Results**

**4.1 Horizontal winds**

The horizontal winds from all four off-zenith lidar directions are shown in Figure 6, as well as the corresponding wind time series from the ECMWF integrated forecast system (ECMWF-IFS) interpolated to the location of the lidar. In the meridional wind, a drastic change during the measurement is observed, which corresponds to small-scale wind variations associated with

the edge of a developing high-pressure region, visible in weather maps of this time period. While at the beginning of the measurement, the meridional winds above 5 km are between 10 m s$^{-1}$ and 20 m s$^{-1}$ (northward), a wind reversal in the





tropopause region is observed, with the winds exceeding -40 m s$^{-1}$ (southward).

In the zonal winds, the previously discussed signal losses are clearly visible as gaps in the top measurement height. Despite the gaps, it was still possible to retrieve zonal wind up to 15 m s$^{-1}$ or higher using the eastward-facing telescope during the

majority of the measurement. Here the wind peak in the tropopause region is also visible at the end of the measurement, with winds reaching 40 m s$^{-1}$.

A first qualitative comparison to the winds from ECMWF-IFS shows the same features in zonal and meridional winds with similar amplitudes. A more quantitative comparison can be found in section 4.3.

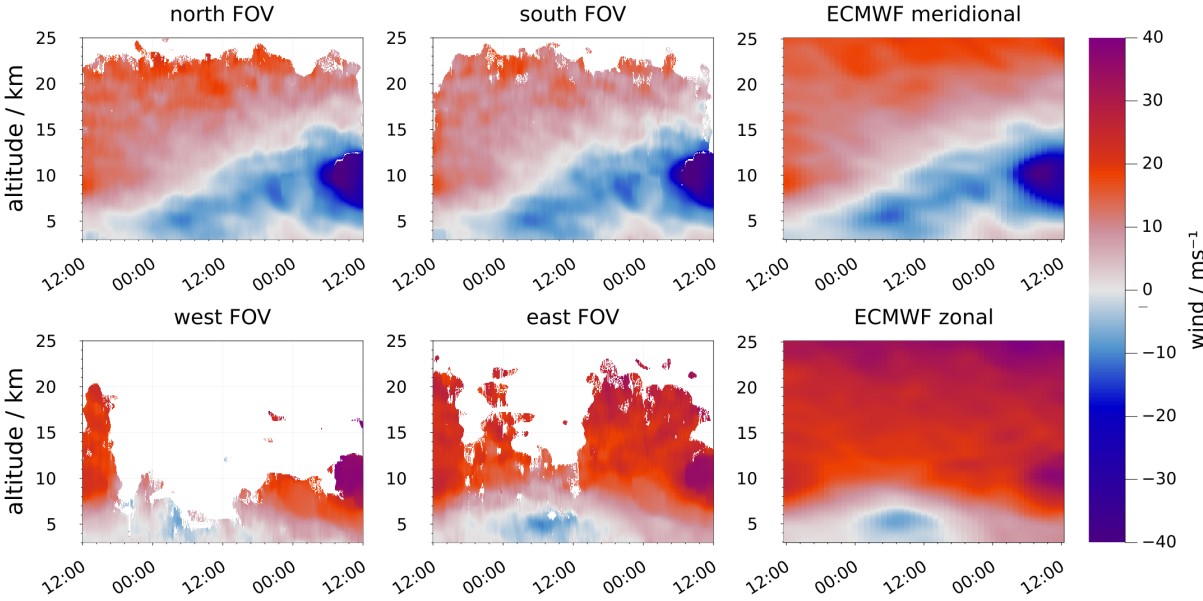

**Figure 6.** Horizontal winds measured between December 16th 12:00 UT and 18th 12:00 UT in the in the north/south/west/east fields of view (see plot titles) using a vertical integration of 1000 m and an integration time of 2 hours. The corresponding meridional and zonal winds from ECMWF are shown in the two right-most panels. The integration window was shifted in 100 m and 2-minute steps. Winds with a line of sight wind error of more than 2 m s$^{-1}$ are masked. The line of sight winds of the tilted telescopes are converted to zonal and meridional winds. In the meridional winds a clear transition around the tropopause from mainly southward winds at the beginning of the measurement to strong northward winds at the end is visible. The lower signal strength in the zonal FOVs causes some gaps in the wind retrieval affecting especially the higher altitudes.

## 4.2 Vertical winds

The vertical winds measured are shown in Figure 7. Overall, the measured vertical winds show absolute wind speeds of 1 m s$^{-1}$ or less, with a clear separation in the behaviour between the troposphere and the stratosphere. While above approximately 11 km the winds fluctuate due to the passage of gravity waves, below a constant background flow of descending air can be observed. A possible influence of the horizontal winds due to pointing errors will be discussed later in Section 5. When comparing the




measured vertical winds with vertical winds from ECMWF, differences are immediately apparent. ECMWF underestimates the

vertical wind fluctuations by approximately an order of magnitude, which is especially visible in the stratosphere. Additionally, ECMWF does not show the same clear separation between the troposphere and stratosphere.

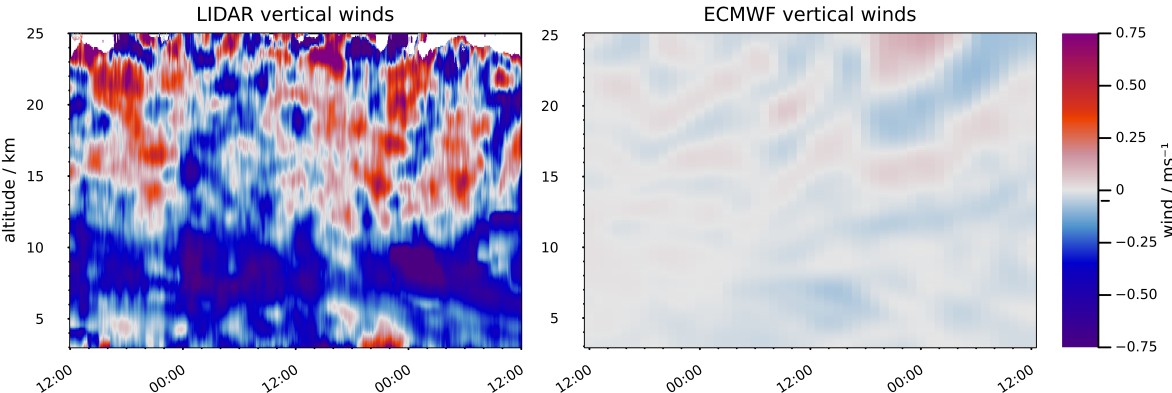

**Figure 7.** Measured vertical winds (left) and the corresponding vertical winds from the ECMWF (right). The vertical winds in ECMWF are underestimated by approximately an order of magnitude. In the upper troposphere a constant background flow of descending air can be observed in the measured winds.

### 4.3   Comparison of lidar winds with ECMWF and Aeolus

In our measurement comparison, we utilized the ECMWF winds in two distinct approaches. Firstly, we directly employed the ECMWF winds for the instrument's location (see Fig. 6, 7, and 8). Secondly, to achieve a more comprehensive comparison

with the individual fields of view, we interpolated and sampled the ECMWF winds along simulated beams (see Fig. 4, 9 and 10). This interpolation and sampling process is very important due to the increasing positional difference between the beams at higher measurement altitudes. At a height of 20 km, the separation between two opposite fields of view reaches 23 km, exceeding the ECMWF horizontal resolution of about 9 km.

    The comparison of the mean winds during the measurement with the time-averaged ECMWF data over the 48 h of the

measurement (see Figure 8) shows good agreement of the ECMWF data with the measurement in the meridional direction.

    The qualitative comparison presented in Fig. 6 shows that the wind pattern observed is similar to the ECMWF zonal wind pattern. In the vertical winds the behaviour discussed in section 4.2 is clearly visible, with downward winds in the lower part of the measurements.

  The comparison of the meridional winds measured along the north and south field of view (see Fig. 9, top left) shows a

systematic difference between the two fields of view, with the winds in the north field of view being larger than in the south field of view. This effect increases with altitude. The column mean of the time-averaged difference is 1.32 m s$^{-1}$ with a standard deviation of 0.88 m s$^{-1}$. The same analysis of the interpolated and beam-sampled ECMWF winds shows a similar





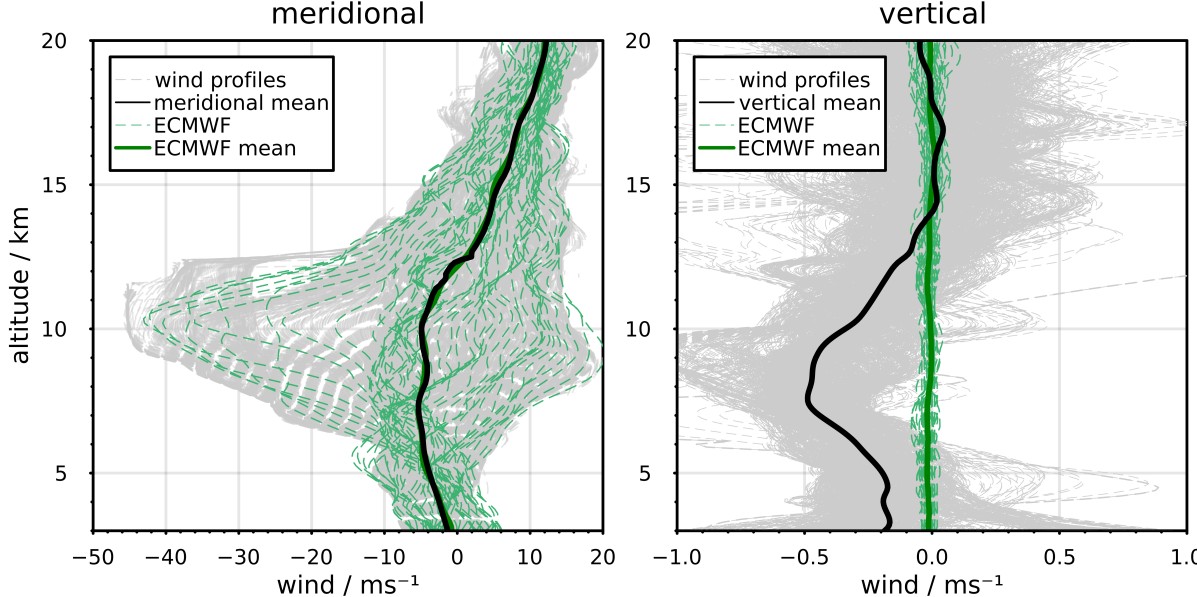

**Figure 8.** Meridional (left) and vertical (right) wind profiles measured (dashed grey lines), together with the calculated mean (thick black line) from 16th 12:00 UT to 18th 12:00 UT, the individual wind profiles from the ECMWF data set (dashed green lines) and the time average ECMWF wind profile (thick green line). The individual wind profiles indicate the variability of the wind during the measurement period, with the highest variability in the meridional winds around the tropopause region.

trend, but but much less pronounced (see Fig. 9, top right), with a column mean of the time-averaged difference of $0.32\ \mathrm{m\ s^{-1}}$ and a standard deviation of $0.36\ \mathrm{m\ s^{-1}}$.

The direct comparison of the time-averaged field of view difference in figure 10 highlights this. In both, the lidar measurements and the ECMWF data, differences of similar altitude dependency are visible, but the difference between the north and south field of view is much more pronounced in the lidar measurements.

The comparison of the meridional winds from the lidar measurement and ECMWF along the individual fields of view (see Fig. 9, bottom) paints a similar picture. Along the north field of view, the meridional winds measured with the lidar are slightly higher on average than the winds in ECMWF but show excellent agreement in general. The column mean of the difference

between lidar and ECMWF winds in the north field of view is $0.30\ \mathrm{m\ s^{-1}}$ with a standard deviation of $0.36\ \mathrm{m\ s^{-1}}$. Along the south field of view, the lidar measured winds are on average lower than the ECMWF winds, with a column average of $-0.93\ \mathrm{m\ s^{-1}}$ with a standard deviation of $0.73\ \mathrm{m\ s^{-1}}$, and differences are more pronounced in the time resolved heatmap. Especially in the stratosphere, the differences show an altitude dependence, associated with the increase in horizontal distance

between the fields of view with altitude. Around the tropopause region, significant deviations are also observable.

For an additional comparison to a Doppler wind lidar, we used measurements from the Aeolus satellite. The closest overpass occurred on 16 December 2022 at 16:40:55 UT over the coordinates (54.615°N, 12.1785°E), so approximately 64 km from the VAHCOLI test site. For the comparison, the VAHCOLI measurements were down-sampled with an altitude resolution of





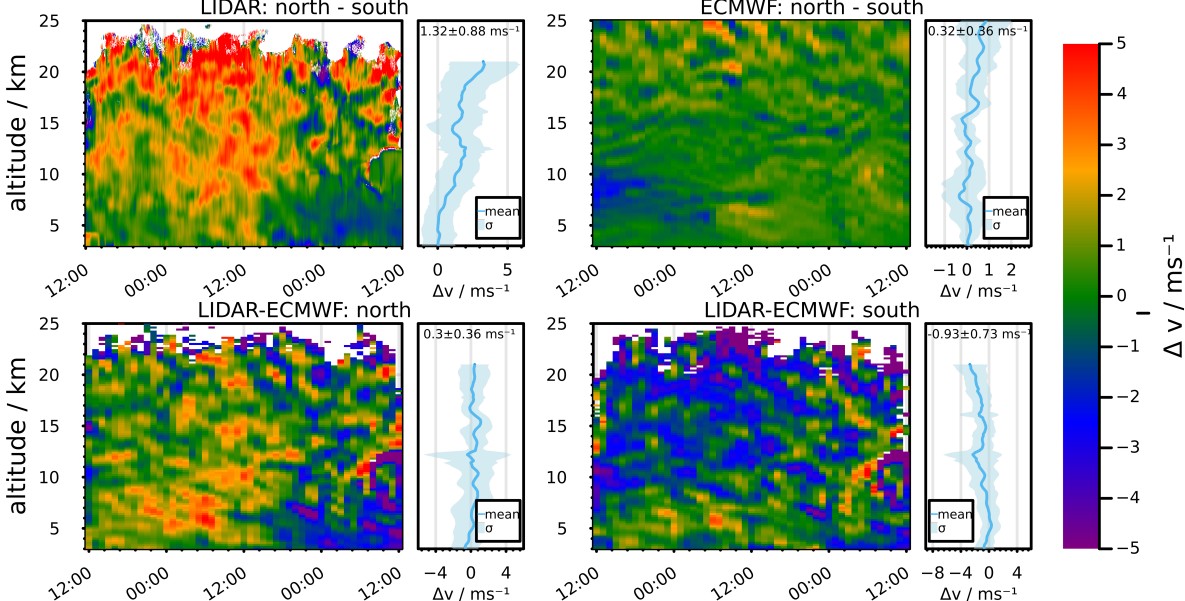

**Figure 9.** Difference of the meridional winds ($\Delta v$) in the time from 16th December 12:00 UT to 18th December 12:00 UT between 3 and 25 km, together with the altitude dependent mean and standard deviation. **top left:** Difference between the north and south field-of-view of the lidar ($\Delta v = v_{\text{lidar,north}} - v_{\text{lidar,south}}$). **top right:** Difference between the ECMWF meridional winds sampled along the north and the south FOV ($\Delta v = v_{\text{ecmwf,north}} - v_{\text{ecmwf,south}}$). **bottom left:** Difference between meridional winds measured along the north FOV of the lidar and the ECMWF winds sampled along the north FOV ($\Delta v = v_{\text{lidar,north}} - v_{\text{ecmwf,north}}$). **bottom right:** Same as bottom left, but for the south FOV ($\Delta v = v_{\text{lidar,south}} - v_{\text{ecmwf,south}}$). The altitude dependant mean and standard deviation has only been calculated up to 21 km when using lidar data, to avoid an influence of gappy data. For every difference the column mean and its standard deviation has been calculated. In the lidar measurement we see a much stronger difference between the north and south field of view than in ECMWF.

1 km by 2 h integration time, to match the altitude resolution of Aeolus. The measured wind components were combined and
projected to the horizontal line of sight of Aeolus. The distance between the Aeolus position and the location of VAHCOLI 1 has been compensated by a time lag of 96 minutes, based on the observed drift of wind features through the multiple field of view of the VAHCOLI instrument. The comparison (see Figure 11) shows good agreement between the projection of the Aeolus HLOS measurement and the projection of the VAHCOLI measurements. Using the 16:40:55 UT profile from Aeolus for statistical analysis of the deviation, like before, results in a standard deviation of 3.31 m s$^{-1}$ and a mean deviation of -0.12 m s$^{-1}$, with
the biggest differences occurring in the stratosphere. The average error estimate of 1.3 m s$^{-1}$ for the projected VAHCOLI wind is by a factor of 2.5 lower than the average error estimate 3.3 m s$^{-1}$ for the Aeolus HLOS wind.





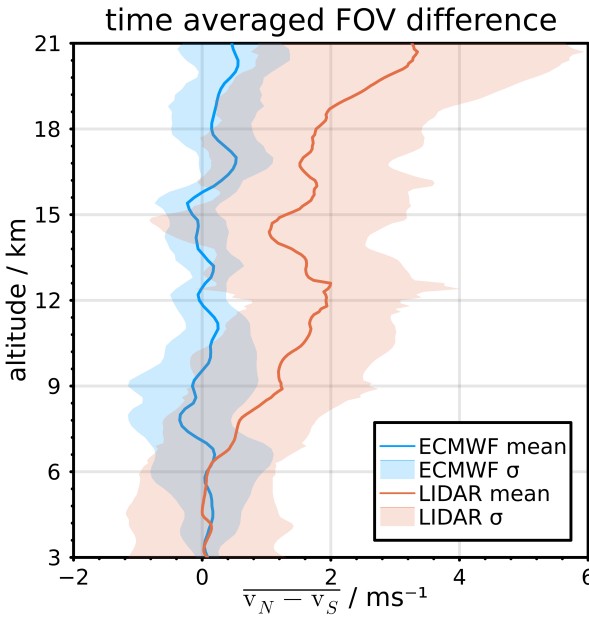

**Figure 10.** Comparison of the time-averaged differences between the north and the south field of view of the lidar measurements and the ECMWF winds sampled along the two fields of view. The time average is taken between the 16th December 12:00 UT and the 18th December 12:00 UT. The blue and orange lines represent the mean difference and the corresponding shading is the standard deviation. The lidar measurement shows a strong altitude dependency which is caused by an asymmetry in the measured meridional winds in the north and south field of view.

## 5 Discussion

The lidar measurements during the period from the 16th December 2022 12:00 UT to the 18th December 2022 12:00 UT, were successful with the system operating for more than 48h. The measured winds in the two meridional fields of view show the same wind pattern. The same is true for the zonal fields of view, despite the gaps in altitude coverage. The vertical pointing accuracy of the system is estimated to be better than 0.5°, based on the internal sensor and the accuracy of the 3D-printed construction. Due to this and the narrow field of view (<30 mrad) we rule out a contamination of the vertical wind by the horizontal wind components. This is supported by the winds in the stratosphere above 13 km, which are zero on average.

Albeit the measured horizontal winds show convincing similarity to the winds reported by ECMWF, both qualitatively and quantitatively, we observe a significant asymmetry of the wind field in meridional direction, which is not well represented in ECMWF. The effect is too large to be caused by a pointing error since for a wind difference of this magnitude between the fields of view a tilting error of 6° is required, which is much larger than our estimated tilting precision of 0.5°. More likely, we observe a true gradient in the wind field above the lidar, associated with the small-scale dynamics at the edge of a developing high-pressure region. ECMWF maps of the meridional wind for the period indicate the presence of such a gradient. We conclude that the ECMWF model is underestimating the gradient in the wind field due to its 9 km grid spacing.



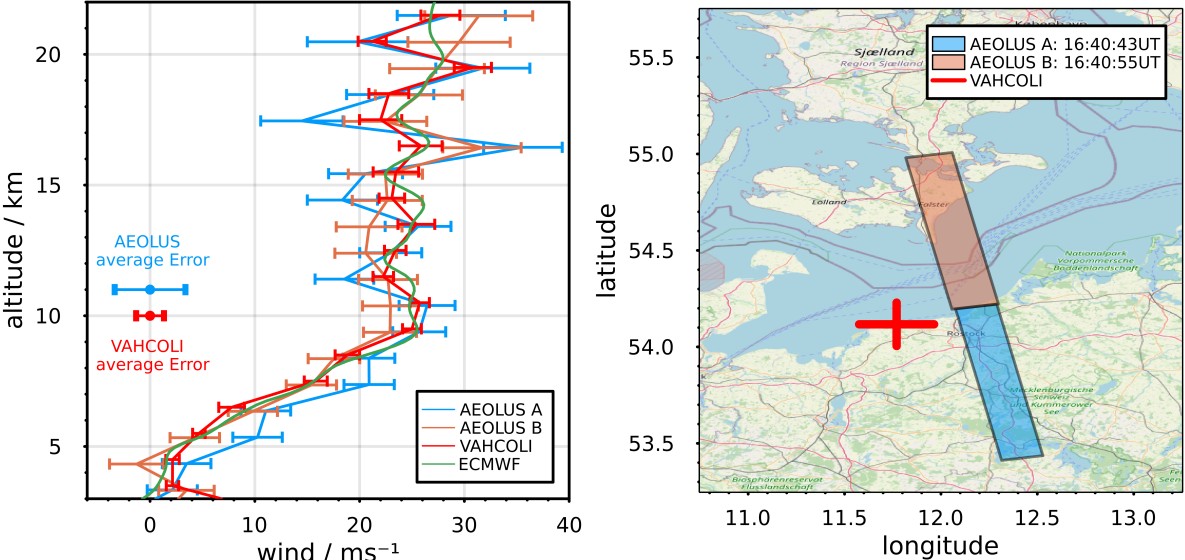

**Figure 11. left:** Comparison of the winds measured by Aeolus (blue and orange lines) and the VAHCOLI 1 lidar system (red lines) together with the winds from ECMWF (green line). For winds from VAHCOLI and ECMWF the meridional, zonal and vertical components have been combined and projected on to the horizontal line of sight (HLOS) of Aeolus. The difference of the lidar measurement and the B profile from Aeolus is $-0.12 \pm 3.31$ m s$^{-1}$ on average, with the best agreement below 15 km. Additionally the average error for the VAHCOLI winds (1.3 m s$^{-1}$, red) and the Aeolus winds (3.3 m s$^{-1}$, blue) are marked. **right:** Map showing the integration areas for the two Aeolus profiles (orange & blue boxes) together with the VAHCOLI beams (red lines) up to a height of 25 km. (data: (© OpenStreetMap contributors, 2017))

This highlights the importance of the multi-FOV Doppler lidar to measure these variations in the transition region between mesoscale and microscale ($10^3 - 10^4$ m horizontally). The assimilation of higher quality, real-time wind measurements will presumably improve the forecast capabilities of high-resolution numerical weather prediction models, by providing information about these typically inaccessible small-scale processes.

Vertically there are significant differences between measured and ECMWF winds, with higher fluctuations and systematic deviations, especially in the troposphere. As discussed before it is not likely that a contamination of the vertical winds by the horizontal winds is the cause for these systematic deviations. Additionally, the negative vertical winds in the troposphere are in agreement with the increase in air pressure measured at ground level, discussed in section 3.1. The larger fluctuations in the vertical wind measurements are expected since lidar measurement volume is much smaller than the model cell size (about

9 km x 9 km in this case) and thus captures smaller scale dynamics that are not captured in the model. The different sizes of the sounding volume and the model volume might also explain other differences between the measured data and ECMWF data. This underestimation of vertical wind speeds in ECMWF has also been previously reported and discussed (Preusse et al., 2014).



Comparison to one profile of Aeolus winds shows a good agreement to our measurements with an average deviation of
$-0.12 \pm 3.31$ m s$^{-1}$. These values are well within the values discussed previously by Martin et al. (2021) for the Aeolus
measurements. In general, the aerosol-based wind profiles from the VAHCOLI 1 instrument cover a larger altitude range than
the Rayleigh winds from Aeolus. The VAHCOLI instruments have been purposefully designed to integrate seamlessly into a
European lidar network in the foreseeable future. In this capacity, they are exceptionally well-suited to complement forthcom-
ing Aeolus successor missions. While the satellites acquire top-down measurements and sparsely sample multiple locations,
the VAHCOLI instruments adopt an Eulerian reference frame, observing the atmosphere from the ground up and offering
continuous measurements. The assimilation of both the global wind measurements from satellites and the ground-based lidar
network measurements into forecast models such as ECMWF promises significant enhancements to their capabilities, owing
to the inherent complementarity of these datasets.

## 6   Conclusions

In this study, we present the very first measurements of winds along five fields covering the UTLS with a lidar. To our knowl-
edge all previous Doppler-wind lidars reaching this altitude used at the most three field of view. The VAHCOLI instrument
with the MFOV upgrade can provide continuous measurements of wind using backscatter from tropospheric and stratospheric
aerosol from 3 up to 25 km in unattended and automated operation, even under challenging weather conditions.

Using the 5 fields of view of the instrument, we are capable of measuring small-scale wind asymmetry associated with the
development of a high-pressure region in real time. We can access the transition region from micro- to mesoscale at horizontal
wavelengths between $10^3$ and $10^4$ m. Assimilation of real-time observations from a network of Doppler-wind lidars with these
capabilities would be useful in improving short-term weather forecast skills by resolving quickly evolving weather features.

We have achieved accurate real-time measurements of vertical wind and have shown that ECMWF underestimates the
vertical wind by an order of magnitude during our case study. Our meridional winds were verified against ECMWF with an
excellent agreement, better than $0.30 \pm 0.36$ m s$^{-1}$ along the north beam of the lidar and $-0.93 \pm 0.73$ m s$^{-1}$ along the south
beam.

We have found a good agreement with a measurement from an Aeolus overpass better than $-0.12 \pm 3.31$ m s$^{-1}$, indicating
that the data reliability of both systems is comparable. We envisage the addition of Rayleigh winds and temperatures to the
lidar unit in the near future.

*Data availability.* VAHCOLI data used in this publication can be found at the following DOI: 10.22000/1720. ECMWF data is freely
available at https://www.ecmwf.int/en/forecasts/access-forecasts/access-archive-datasets. Aeolus data is freely available at https://aeolus-
ds.eo.esa.int/oads/access/. Map data is freely available from https://openstreetmap.de/karte/.



*Author contributions.* T.H.M., J.H, J.F., A.Ma. built the VAHCOLI 1 instrument. T.H.M developed and built the MFOV upgrade, wrote the code for the analysis and drafted the manuscript. J.H. wrote the operating system of the instrument, developed the measurement technique and provided supervision. G.B. provided the interpolated ECMWF data and provided supervision. J.F. developed spectral methods. A.Ma. helped with the design of the MFOV upgrade. A.Mu. built the laser. R.W. helped with the Aeolus data and provided supervision. F.-J.L. provided supervision and scientific insight. All Authors contributed to editing the manuscript.

*Competing interests.* Robin Wing and Gerd Baumgarten are members of the editorial board of Atmospheric Measurement Techniques.

*Acknowledgements.* We thank Corwin Wright and Timothy Banyard from University of Bath, Uk for providing the Aeolus data.

We thank Sarah Scheuer and Michael Strotkamp from the Fraunhofer Institute for Laser Technology ILT, Aachen, Germany for their substantial contributions to the development of the diode-pumped alexandrite ring laser and the helpful discussions.

Map data copyrighted OpenStreetMap contributors and available from https://www.openstreetmap.org

This work is partially supported by the Leibniz SAW project FORMOSA: FOur-dimensional Research applying Modelling and Observations for the Sea and the Atmosphere (grant no. K227/2019). Further support was provided by the Collaborative Research Centre TRR 181 "Energy Transfers in Atmosphere and Ocean" funded by Deutsche Forschungsgemeinschaft (DFG, German Research Foundation) - Projektnummer 274762653 and the project Analyzing the Motion of the Middle Atmosphere Using Nighttime RMR-lidar Observations at the Midlatitude Station Kühlungsborn (AMUN) funded by Deutsche Forschungsgemeinschaft (DFG) - Projektnummer 445400792.



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
