# Peer review of "3D-Wind Observations with a Compact Mobile Lidar based on Tropo- and Stratospheric Aerosol Backscatter"

_EGUsphere, 2023_

## Referee Comment (RC1)

04.10.2023

Review of the article

**"First measurements of 3-Dimensional winds up to 25 km based on Aerosol backscatter using a compact Doppler lidar with multiple fields of view "**

submitted by Mense et al.
**(AMT)**

**Review Summary**

The authors introduce the first 3D wind measurements up to the stratosphere based on an updated version of their VAHCOLI system. In particular, they extended the system by multiple telescopes that enable the simultaneous measurements of 4 LOS directions (N, S, W, E) as well as the vertical component. They demonstrate the functionality of the system based on an exemplary measurement and comparison to both ECMWF data as well as Aeolus data. The paper is well-structured and clearly understandable. Still, a few important pieces of information to interpret the presented results are missing. Furthermore, a few of the drawn conclusions are questionable. Hence, I recommend accepting the paper for publication in AMT after major revisions based on the comments that are listed below.

**General comments**

- Although it is appreciated that the VAHCOLI system is introduced in L Lübken and Höffner, 2021, it would be very helpful to recapitulate the overall measurement principle and the basic specifications, such that the presented results can be interpreted without going to other references.

- The term "ADM" was omitted by ESA several years ago. Since then, it is just Aeolus. This should be adapted throughout the manuscript.

- The paper title is a bit misleading, as it says that 3D wind measurements are demonstrated up to 25 km altitude, which is not true as only the north – south beams reach the 25 km, whereas this is not true for the east – west beam which only reach lower altitudes.

**Detailed comments**

- Abstract: A few acronyms are not introduced in the abstract (VAHCOLI, ECMWF)

- Abstract: You mention an excellent agreement between VAHCOLI and ECMWF data. Although I agree that a random error of < than 1 m/s is rather good, a systematic error of -0.93 m/s is not. Hence, I suggest to adapt this sentence.

- Abstract, comparison to Aeolus: At this stage, it is not clear what you compare here. For the ECMWF comparison, it is the HLOS for the north and south beam, but for Aeolus, it is likely to be a projection to the Aeolus viewing direction. This should be clarified here.

- Line 39, Aeolus error estimates: It should be emphasized that both the systematic but even more the random error of Aeolus Rayleigh clear winds depend on the actual signal levels and can reach values up to 7.5 m/s. Due to the decreasing signal levels throughout the mission, the overall random error increased. But also, in for instance dust-laden areas with lower signal levels, the random error can be significantly higher. This is for instance shown in Witschas et al. 2022, Fig. 8 by means of a statistical comparison to airborne wind lidar data. It

would be worth mentioning here and adding further references related to Aeolus wind product validation.

- Line 60, unique possibility to measure wind based on aerosol: It is not understood why you argue that this is a unique possibility. For instance, the ALADIN airborne demonstrator A2D measures wind with a similar technique to Aeolus and a rather good sensitivity due to the larger signal levels. Further, Ball Aerospace developed a wind lidar system based on a Mach-Zehnder (OAWL) interferometer that is very sensitive to aerosol return (Baider, 2018, The Optical Autocovariance Wind Lidar, Part II: Green OAWL (GrOAWL) Airborne Performance and Validation). Last but not least, heterodyne detection wind lidar systems demonstrated the ability to measure wind speed also in the vicinity of very weak aerosol concentrations. This should be appreciated here.

- Fig. 1: It would be helpful to indicate the respective parts (for instance the camera window) in the picture.

- Line 77, new approach: It is not agreed that the approach of using multiple FOV is really new. This was done in several other wind lidars before, for instance, "Narasimha S. Prasad, 2017 - Three-beam aerosol backscatter correlation lidar for wind profiling". This should be appreciated.

- Line 97: You mention the simultaneous measurement of u,v,w. This would also allow to directly derive the momentum flux from the data. Have you verified if this works properly?

- Line 111, carefully selected materials: Which are the materials that you selected, and why?

- Line 113, pointing precision: You mention that the pointing precision is only limited by the accuracy of the 3D print, but it is not mentioned how accurate the 3D printing is. Please adapt this information and mention the estimated pointing precision that results from this uncertainty.

- Line 140, icing problem and operation time: You mention that you had an icing problem during the discussed measurements but also that you have a further 600 hours of measurement data available. Hence, I was wondering if it would make sense to add a measurement case without an icing problem to demonstrate the maximum range the system can reach. I guess this would be a rather valuable extension.

- Line 143, Mie channel is used: Does the VAHCOLI system also has a Rayleigh channel? How far do you expect to reach with it?

- Fig. 2: You mention that you apply a running average of 2 hours in the horizontal and 1 km in the vertical, but the magenta area (right, top) shows features with a higher resolution. How is this possible? I would expect that these features smear out due to the averaging. Considering the moving average, what is the actual horizontal and vertical resolution of your data?

- Line 152: Fig 5 is mentioned and discussed before Fig. 4 is addressed. Maybe this should be rearranged.

- Line 155, Gaussian distribution: Here it would be worth mentioning that the spectral shape differs from a pure Gaussian in the lower troposphere (Rayleigh-Brillouin scattering) which will not impact your measurement results.

- Fig. 4, caption: How do you define your SNR? Further, you first mention the abbreviation SNR, before you explain it afterwards.

- Fig. 4, 5[th] line: top left should be bottom left.

- Fig. 4, caption, corresponding ECMWF winds: Do you interpolate ECMWF data to your measurement location, or do you use a next-neighbor approach which could also explain the discrepancies? Would be worth adding this detail. You write that in a later section, but the information should already be provided here.

- Fig. 5: freqency → frequency

- Line 206, separation of troposphere and stratosphere: It would be worth adapting the color-scale of the ECMWF data plot (e.g. by a factor of 10). With that, any features could be more easily recognized.

- Fig. 8: In my opinion, it would be interesting to see a scatter plot of ECMWF vs. VAHCOLI winds. The color of the data points could additionally indicate the altitude. By doing so, it could be demonstrated if there is for instance a systematic difference between the two data sets and how large the random error is.

- Line 223: but but, delete one

- Line 234, height-dependent bias: What is your explanation for the high-dependent bias? The discrepancy between the two beams? Isn't that much less important than the long averaging time? Do you also have radiosonde profiles that could be used for comparison? It would be interesting to clarify if there is an issue with the VAHCOLI wind retrieval or with the used reference data sets, e.g. ECMWF.

- Line 236, ff: As Aeolus data is assimilated in the ECMWF model, these two data sets are not really independent. This fact should be highlighted somewhere. Furthermore, it is important to add information on the Aeolus processor baseline that has been used. The Aeolus processor experienced a lot of modifications throughout the mission time, which also had an impact on the Aeolus data quality. In addition, it is not clear if you are using Rayleigh-clear winds or Mie-cloudy winds for your comparison.

- Line 239, 1 km integration: Usually, the Aeolus range bin size is variable. Do you take this into account, or do you "just" average to 1 km range gates?

- Line 241, drift correction: How large is this drift correction? Would be interesting to see the wind profiles before and after correction to get an idea about the actual impact of this correction procedure.

- Line 245, biggest differences in the stratosphere: Is this in line with the estimated error that is also reported in the Aeolus data product? As the signal levels for Ray-clear winds are lowest in the stratosphere, the estimated error should be large - as observed.

- Line 260: You conclude that the ECMWF model is underestimating wind gradients. Can you be really sure that the beam separation, the spatial and temporal discrepancy between lidar measurements and model, and the 2-hour average cannot introduce such discrepancies? Would be worth excluding all other potential error sources and adding for instance a radiosonde comparison that supports this statement.

- Fig. 11: Is this Aeolus Rayleigh-clear or Mie-cloudy winds or a mixture? It is not understood what you are comparing here.

- Line 263, Assimilation of high-quality winds: Are you referring to winds measured from ground-based lidar systems in the framework of a network here, or to upcoming spaceborne wind lidars as for instance Aeolus 2?

- Line 278: What is the overall goal of a wind lidar network?

---

## Author Comment (AC1)

**(Author response)**

Original Comment:

In the introduction, the importance of wind field measurement, the methods of wind field measurement and the shortcomings of various methods are introduced in detail, which leads to the development progress of coherent lidar. And explains the reasons why spaceborne lidar need to use both Mie and Rayleigh lidar, thus leading to the technical progressiveness of this manuscript. Shows the capable of detecting Mie signals that better than ALADIN.

In line 115, the angle correction methods is very good, effectively avoiding wind field errors caused by angle deviation.

Thank you. Quite some effort went into ensuring the correct alignment of the instrument.

In line 145, 'For data processing the individual pulse raw dat…', seems syntax error.

Thank you, but we don't think there is a syntax error here. The system measures each individual laser pulse and the laser has a repetition rate of 500Hz, thus we get a pulse to pulse time resolution of 2ms. The electronics run with 150Mhz giving an altitude resolution of 1m.

We added in line 148:"…electronic altitude resolution of 1m,…"

Above 150, eight resolution of 1 m for each individual field of view. What is the width of the laser pulse?

The length of the laser pulse is 1.1 microseconds temporally (FWHM) and thus roughly 165m in length.

We added in line 149:" The laser pulse itself has a length of approximately 165m (full width at half maximum) in the atmosphere."

In line 205, obviously, it can be used to correct ECMWF, but the transmission of gravity waves above 11km is only a guess, and a simple wavelet analysis can be done for simple analysis.

We changed in line 202:" While above 11~km the winds fluctuate around zero, presumably due to the passage of gravity waves, below a constant background flow of descending air can be observed."

A closer analysis of the fluctuations in the vertical winds are planned for our future work. At this point of the work you are correct, that the attribution of the fluctuation to gravity waves is only a guess, albeit a reasonable one given the shape and temporal evolution of the fluctuation seen in figure 7.

In figure 11. The satellite and lidar are several tens of kilometers apart, and this comparison result is still very good, can you explain the reason?

During the time of the Aeolus overpass there was no quick change in wind was observed, both in ECMWF and in our lidar measurements. We thus assume that the wind field was quite homogenous.

---

## Author Comment (AC2)

*Very interesting paper. However, we at ECMWF would like some clarifications regarding the use of the ECMWF vertical wind component, which is compared to the lidar observations.*

*Vertical wind is not something we would expect the ECMWF IFS (normal set-up) to capture well, especially in the hydrostatic model that diagnoses it from vertical derivative of the horizontal divergence of the wind field.*

*Could you confirm how you calculate the vertical wind in [m/s] as we provide it in MARS as omega (so in units of [Pa/s]). Do you use the hydrostatic approximation to say w=-omega/(rho\*g)?*

*And most importantly, do you make sure that you retrieve the full resolution data from MARS at 9 km rather than the truncated field (this is a common mistake by many). For horizontal wind this should not matter too much as it has a lot of power in low wavenumbers, but for the vertical winds this will most likely lead to a severe under-estimation as most of the power is in large wavenumbers that will be ignored by truncation.*

*Kind regards*

Thank you for looking into this, we are very happy to share details that we may want to include in a revised manuscript.

*Could you confirm how you calculate the vertical wind in [m/s] as we provide it in MARS as omega (so in units of [Pa/s]). Do you use the hydrostatic approximation to say w=-omega/(rho\*g)?*
**Yes we confirm that w is calculated as you describe.**

**In Detail:**

```
modlevel = ncv["level"][:]
surface_geop = ncv["z"][:][0, 0, IDX, :]
lnsp = ncv["lnsp"][:][tstep, 0, IDX, :]
T = ncv["t"][tstep, :, IDX, :]
Q = ncv["q"][tstep, :, IDX, :]
wpress = ncv["w"][tstep, :, IDX, :]

plevel, phlevel = calc_ECMWF_press_3d(lnsp, modlevel=modlevel)

geop = calc_ECMWF_geop_3d(phlevel, T, Q, surface_geop)
Tv = T_virtual(T, Q)

z, g = calc_ECMWF_altitude_g_3d(geop, latitude)

rho = calc_density(plevel, Tv)

w = wpress / (-rho * g)
```

**Where „wpress" is „omega" in [Pa/s]**

*And most importantly, do you make sure that you retrieve the full resolution data from MARS at 9 km rather than the truncated field:*

**We specify the full spectral resolution with „res=1279" statement.**
However we noticed that the keyword may have changed to „resol"
https://confluence.ecmwf.int/pages/viewpage.action?pageId=171422484

**Please find below the complete retrieve statements:**

```
  server = ecmwfapi.ECMWFService("mars")
…
  basepar={
       "stream"  : "oper",
       "class"   : "od",
       "type"    : "an",
       "expver"  : "1",
       "date"    : "{}".format(date),
       "time"    : "0",
       "grid"    : "0.25/0.25",
       "levtype" : "ml",
       "levelist": "1",
       "area"    : "{}/{}/{}/{}".format(latlim[1], lonlim[0], latlim[0], lonlim[1]),
       "domain"  : "A",
       "res"     : "1279",
       "param"   : "129.128",
       }
  b=basepar.copy()

  execute(server, basepar, filekey + "_1.grb")

  b["param"]="152.128"
  b["type"]="fc"
  b["time"]="0/12"
  b["step"]="0/1/2/3/4/5/6/7/8/9/10/11"

  execute(server, b, filekey + "_2.grb")

  b["param"]="T/U/V/W/O3/155.128/138.128/133.128/248.128"
  b["levelist"]="1/to/{}".format(levno)

  execute(server, b, filekey + "_3.grb")

  combine_forecast_modlev(filekey)
```

We like to mention that the downsampling to the 0.25° grid (to save space) lead to sligtly uncertain results.
We repeated the retrieval and sampled on a 0.125/0.0625 grid and observed differences of less than 0.1 m/s in the vertical wind.

[Figure]

Fig. 1: Vertical wind differences of data sampled on a 0.25/0.25 grid minus 0.125/0.0625 grid.

For the meridional wind gradient we find most often differences of less than 1 m/s

[Figure]

Fig. 2: Meridional wind gradient differences of data sampled on a 0.25/0.25 grid minus 0.125/0.0625 grid.

We can also confirm, that the effect on the horizontal winds and especially on the observation of the wind gradient is not affected by this.

[Figure]

Fig. 3: Same as Figure 9 in the Paper, but with the updated ECMWF retrieval. No significant difference is visible.

During our investigation of Figure 9, we realised, that a software error caused the observational filter to not be correctly applied. We corrected this, leading to a slightly changed figure 9. The conclusions drawn from the figure remain the same.

[Figure]

Fig. 4: Updated figure 9, with corrected observational filtering applied to the ECMWF data.

---

## Author Comment (AC3)

**(Author response)**

**Review Summary**

The authors introduce the first 3D wind measurements up to the stratosphere based on an updated version of their VAHCOLI system. In particular, they extended the system by multiple telescopes that enable the simultaneous measurements of 4 LOS directions (N, S, W, E) as well as the vertical component. They demonstrate the functionality of the system based on an exemplary measurement and comparison to both ECMWF data as well as Aeolus data. The paper is well-structured and clearly understandable. Still, a few important pieces of information to interpret the presented results are missing. Furthermore, a few of the drawn conclusions are questionable. Hence, I recommend accepting the paper for publication in AMT after major revisions based on the comments that are listed below.

**General comments (GC)**

GC1: Although it is appreciated that the VAHCOLI system is introduced in L Lübken and Höffner, 2021, it would be very helpful to recapitulate the overall measurement principle and the basic specifications, such that the presented results can be interpreted without going to other references.

We have added the following text to line 91:

"Based on this hardware the lidar can do high resolution spectroscopy of the backscattered signal. For this the frequency of the power laser is quickly and precisely scanned over two highly stabilised filters, a broadband solid etalon with an FWHM close to the Doppler width of the atmospheric molecular line and a narrowband confocal etalon (FWHM 7.5 MHz). Of the two APDs, one records the signal, which is transmitted through the planar and confocal etalon (APD1) and the other one records the signal, which is transmitted through the planar etalon but is reflected on the confocal etalon (APD2). The resulting spectra contain a mixture of Mie and Rayleigh backscatter on APD2 and mostly Mie backscatter on APD1 with a little bit (roughly 1/60th) of Rayleigh backscatter as background. We derive atmospheric properties like wind and aerosol content from these measured spectra."

In line 92 we added another reference for the alexandrite laser, which was recently published: (Munk et al., 2023)

In line 137 we added: "During this measurement period, the planar etalon discussed in section 2.1 was not installed, leading to a slight increase in the daylight background. Despite this, no significant performance drops connected to this daylight background were observed."

GC2: The term "ADM" was omitted by ESA several years ago. Since then, it is just Aeolus. This should be adapted throughout the manuscript.

We have removed "ADM" throughout the manuscript

GC3: The paper title is a bit misleading, as it says that 3D wind measurements are demonstrated up to 25 km altitude, which is not true as only the north – south beams reach the 25 km, whereas this is not true for the east – west beam which only reach lower altitudes.

We have changed the title to "3D-Wind Observations with a Compact Mobile Lidar based on Tropo- and Stratospheric Aerosol Backscatter"

In line 4 for we changed the sentence to read: "The method was applied at the edge of a developing high-pressure region, covering altitudes between 3 and 25~km."

**Detailed comments (DC)**

DC1: Abstract: A few acronyms are not introduced in the abstract (VAHCOLI, ECMWF)

We have changed line 1 to read:

"We present the first measurements of simultaneous horizontal and vertical winds using a new lidar system developed at the Leibniz Institute of Atmospheric Physics in Kühlungsborn, Germany (54.12°N, 11.77°E), for the concept of Vertical and Horizontal coverage by lidar (VAHCOLI)."

We have added to line 5:

"European Centre for Medium-Range Weather Forecasts (ECMWF)"

DC2: Abstract: You mention an excellent agreement between VAHCOLI and ECMWF data. Although I agree that a random error of < than 1 m/s is rather good, a systematic error of -0.93 m/s is not. Hence, I suggest to adapt this sentence.

We have rephrased line 5 in the abstract to read: "Comparisons between the lidar measurements and data from the European Centre for Medium-Range Weather Forecasts (ECMWF) show excellent agreement for the meridional wind component along the north beam of the lidar, better than 0.30 ± 0.33 m s−1, while along the south beam a higher deviation with −0.93 ± 0.73 m s−1 is observed. "

DC3: Abstract: Comparison to Aeolus: At this stage, it is not clear what you compare here. For the ECMWF comparison, it is the HLOS for the north and south beam, but for Aeolus, it is likely to be a projection to the Aeolus viewing direction. This should be clarified here.

Since we observe the vertical component of the wind and LOS wind of the tilted telescopes, we can derive the meridional and horizontal component of the wind.

We have rephrased in line 5 "horizontal component" to "meridional component"

For the comparison to Aeolus all 3 measured wind components are projected to the Aeolus HLOS vector.

We have changed in line 7: "Comparison of Aeolus winds to the lidar winds projected to the Aeolus viewing direction shows…"

DC 4: Line 39, Aeolus error estimates: It should be emphasized that both the systematic but even more the random error of Aeolus Rayleigh clear winds depend on the actual signal levels and can

reach values up to 7.5 m/s. Due to the decreasing signal levels throughout the mission, the overall random error increased. But also, in for instance dust-laden areas with lower signal levels, the random error can be significantly higher. This is for instance shown in Witschas et al. 2022, Fig. 8 by means of a statistical comparison to airborne wind lidar data. It would be worth mentioning here and adding further references related to Aeolus wind product validation

Thank you, we fully agree.

We have added the following text in line 40, and cited witschas et al., 2022 and Ratynski et al., 2023: "Later validation campaigns showed even higher random errors due to the decreasing signal levels throughout the mission. Witschas et al. (2022) report a random error for Rayleigh clear winds of 5.5 m s−1 to 7.1 m s−1 in the altitude range from 0.5-10.5 km and Ratynski et al. (2023) report an error between 5.37 m s−1 and 6.49 m s−1 in the altitude range up to 27 km."

DC5: Line 60, unique possibility to measure wind based on aerosol: It is not understood why you argue that this is a unique possibility. For instance, the ALADIN airborne demonstrator A2D measures wind with a similar technique to Aeolus and a rather good sensitivity due to the larger signal levels. Further, Ball Aerospace developed a wind lidar system based on a Mach- Zehnder (OAWL) interferometer that is very sensitive to aerosol return (Baider, 2018, The Optical Autocovariance Wind Lidar, Part II: Green OAWL (GrOAWL) Airborne Performance and Validation). Last but not least, heterodyne detection wind lidar systems demonstrated the ability to measure wind speed also in the vicinity of very weak aerosol concentrations. This should be appreciated here.

Thank you, we have added the references to appreciate the contribution.

We have changed line 59, ff to read: "Other wind lidar instruments, like the Aladin A2D Airborne Demonstrator, Ball Aerospaces airborne Green OAWL or heterodyne detection wind lidars in general, use Mie scattering to measure wind speeds covering mostly the troposphere (Lux et al., 2022; Baidar et al., 2018). However, as will be shown in the following sections, the lidar systems introduced in this work have the capacity to detect aerosols up to 25km that are invisible to ALADIN and use them for precise wind measurements. This capability is unique for ground based instruments and makes a cluster of them especially interesting for intercomparison studies to upcoming Aeolus successors."

DC6: Fig. 1: It would be helpful to indicate the respective parts (for instance the camera window) in the picture.

We have updated figure 5.

DC7: Line 77, new approach: It is not agreed that the approach of using multiple FOV is really new. This was done in several other wind lidars before, for instance, "Narasimha S. Prasad, 2017 - Three-beam aerosol backscatter correlation lidar for wind profiling". This should be appreciated

We removed "new" from line 77.

DC8: Line 97: You mention the simultaneous measurement of u,v,w. This would also allow to directly derive the momentum flux from the data. Have you verified if this works properly?

Yes, we intend to derive momentum flux in the future. It has not yet been attempted.

DC9: Line 111, carefully selected materials: Which are the materials that you selected, and why?

After some testing, we found that printing the sensitive parts out of white PETG proved to solve the issue in combination with the cooling solution.

DC10: Line 113, pointing precision: You mention that the pointing precision is only limited by the accuracy of the 3D print, but it is not mentioned how accurate the 3D printing is. Please adapt this information and mention the estimated pointing precision that results from this uncertainty.

For our printing methods the dimensional accuracy is better than 1mm per 1m and with surface imperfections of less than 0.3mm. From this and the CAD drawings we can estimate the angular accuracy with which the telescopes are held to be better then 2mrad.

We have added in line 114:" From the dimensional accuracy and surface quality our 3D printers produce, we estimate the angular error introduced by the 3D printed construction to be less than 2mrad."

DC11: Line 140, icing problem and operation time: You mention that you had an icing problem during the discussed measurements but also that you have a further 600 hours of measurement data available. Hence, I was wondering if it would make sense to add a measurement case without an icing problem to demonstrate the maximum range the system can reach. I guess this would be a rather valuable extension.

Most of the 600h of measurements were not done with all five fields of view since other aspects of the system were of higher importance for the further development of the instrument. We thus limit the presented results here to this interesting case during December 2022.

We added a clarifying statement in line 140:"… 600 hours of measurements , of which 85 hours were done in MFOV mode."

DC12: Line 143, Mie channel is used: Does the VAHCOLI system also has a Rayleigh channel? How far do you expect to reach with it?

The VAHCOLI System has a Rayleigh channel. In the current version it is not used for Doppler measurements, but will be in the future. We expect to reach altitudes around 50km with it.

DC13: Fig. 2: You mention that you apply a running average of 2 hours in the horizontal and 1 km in the vertical, but the magenta area (right, top) shows features with a higher resolution. How is this possible? I would expect that these features smear out due to the averaging. Considering the moving average, what is the actual horizontal and vertical resolution of your data?

We shift the integration window of 2h and 1km in 2 minute and 100m steps.

We clarified in the caption: " The data is smoothed by the 2 hours and 1km integration window which is shifted in 2 minutes and 100m steps. The data is then converted to the unit of counts per time per altitude."

DC14: Line 152: Fig 5 is mentioned and discussed before Fig. 4 is addressed. Maybe this should be rearranged.

We swapped the order of Fig. 4 and Fig. 5.

DC15: Line 155, Gaussian distribution: Here it would be worth mentioning that the spectral shape differs from a pure Gaussian in the lower troposphere (Rayleigh-Brillouin scattering) which will not impact your measurement results.

To clarify, we changed in line 151: "Winds are derived from the Mie backscatter spectra by fitting a Voigt-function to each height channel, with an example of this fit shown in Figure 4."

DC16: Fig. 4, caption: How do you define your SNR? Further, you first mention the abbreviation SNR, before you explain it afterwards.

We added to the caption of Figure 4. (previously Fig. 5, but change due to DC14): "… Signal-to-noise ratios (SN R = PeakSignal/Background)…"

DC17: Fig. 4, 5th line: top left should be bottom left.

We have corrected this in the caption of Fig. 5 (previously Fig. 4).

DC18: Fig. 4, caption, corresponding ECMWF winds: Do you interpolate ECMWF data to your measurement location, or do you use a next-neighbour approach which could also explain the discrepancies? Would be worth adding this detail. You write that in a later section, but the information should already be provided here.

We interpolate to our measurement location.

After line 184 we added a section describing how Aeolus and ECMWF data is handled, reading:

" 3.3 ECMWF and Aeolus winds

In this work we utilized the ECMWF winds in two distinct approaches. Firstly, we directly employed the ECMWF winds interpolated to the instrument's location (see Fig. 6, 7, and 8). Secondly, to achieve a more comprehensive comparison with the individual fields of view, we interpolated and sampled the ECMWF winds along simulated beams (see Fig. 5, 9 and 10). This interpolation and sampling process is very important due to the increasing positional difference between the beams at higher measurement altitudes. At a height of 20 km, the separation between two opposite fields of view reaches 23 km, exceeding the ECMWF horizontal resolution of about 9 km.

The Aeolus data has been reprocessed using the baseline 2B15 and regridded to an altitude resolution of 1 km, as in Banyard et al. (2021). Only Rayleigh-clear winds are used due to the scarcity of Mie-cloudy winds. It is worth noting here, that Aeolus winds and ECMWF winds are not really independent, since Aeolus data was assimilated into the ECMWF model."

We removed the information now contained in this section from section 4.3., line 208, ff.

DC19: Fig. 5: freqency → frequency

Done

DC20: Line 206, separation of troposphere and stratosphere: It would be worth adapting the color-scale of the ECMWF data plot (e.g. by a factor of 10). With that, any features could be more easily recognized.

Done

DC21: Fig. 8: In my opinion, it would be interesting to see a scatter plot of ECMWF vs. VAHCOLI winds. The color of the data points could additionally indicate the altitude. By doing so, it could be demonstrated if there is for instance a systematic difference between the two data sets and how large the random error is.

We had a look at the scatter plot and don't see any benefit from including it in the publication. It provides no additional information compared to Figure 9.

[Figure]

DC22: Line 223: but but, delete one

Done

DC23: height-dependent bias: What is your explanation for the high-dependent bias? The discrepancy between the two beams? Isn't that much less important than the long averaging time? Do you also have radiosonde profiles that could be used for comparison? It would be interesting to clarify if there is an issue with the VAHCOLI wind retrieval or with the used reference data sets, e.g. ECMWF

Yes, we assume the horizontal distance between the two beams, which increases with altitude, together with the measurement at the edge of a developing high-pressure region to be the explanation of the height dependant bias. Sadly no radiosonde measurements are available to us during the time of our measurement which are in sufficient vicinity to our field of view.

DC24: Line 236, ff: As Aeolus data is assimilated in the ECMWF model, these two data sets are not really independent. This fact should be highlighted somewhere. Furthermore, it is important to add information on the Aeolus processor baseline that has been used. The Aeolus processor experienced a lot of modifications throughout the mission time, which also had an impact on the Aeolus data quality. In addition, it is not clear if you are using Rayleigh-clear winds or Mie-cloudy winds for your comparison.

We added this to the new section 3.3. Please see the answer to DC18.

DC25: Line 239, 1 km integration: Usually, the Aeolus range bin size is variable. Do you take this into account, or do you "just" average to 1 km range gates?

We added this to the new section 3.3. Please see the answer to DC18.

DC26: Line 241, drift correction: How large is this drift correction? Would be interesting to see the wind profiles before and after correction to get an idea about the actual impact of this correction procedure.

On average the drift correction is -0.75 m/s. Here is an uncorrected figure 11:

[Figure]

DC27: Line 245, biggest differences in the stratosphere: Is this in line with the estimated error that is also reported in the Aeolus data product? As the signal levels for Ray-clear winds are lowest in the stratosphere, the estimated error should be large - as observed.

Yes, this is in line with the increased estimated error in the stratosphere.

We changed in line 244,ff:"…and a mean deviation of -0.12 m s−1. As in line with the stratospheric increase of the error estimate reported in the Aeolus data product, the biggest differences occur in the stratosphere. "

DC28: Line 260: You conclude that the ECMWF model is underestimating wind gradients. Can you be really sure that the beam separation, the spatial and temporal discrepancy between lidar measurements and model, and the 2-hour average cannot introduce such discrepancies? Would be worth excluding all other potential error sources and adding for instance a radiosonde comparison that supports this statement.

The beam separation is the very reason, why we are able to observe the gradient at all. The only way we would see the gradient in our measurement, without a real gradient in the atmosphere, is a massive discrepancy between the viewing angles of the telescopes of 6° or more as discussed in Line 256, ff. This is ruled out by the construction of the instrument and the measures we took to ensure proper alignment.  To exclude effects due to spatial and temporal differences between observation and model we interpolated the model output and sampled it to match exactly the location of the lidar beams in the sky. We then did the analysis with and without putting an observational filter on the ECMWF data and saw no difference in the conclusions we could draw. By this we made sure to exclude the most probable error sources. As in the answer to DC23 stated no such radiosonde measurements are available.

DC29: Fig. 11: Is this Aeolus Rayleigh-clear or Mie-cloudy winds or a mixture? It is not understood what you are comparing here.

We used Rayleigh-clear winds only.

We added to the caption of figure 11:"Comparison of Rayleigh-clear winds measured by Aeolus … and Mie winds measured by the VACOLI 1…"

DC30: Line 263, Assimilation of high-quality winds: Are you referring to winds measured from ground-based lidar systems in the framework of a network here, or to upcoming spaceborne wind lidars as for instance Aeolus 2?

We changed Line 262,ff to read:" The assimilation of higher quality, real-time wind measurements, as provided by a network of ground-based lidar systems,…"

DC31: Line 278: What is the overall goal of a wind lidar network?

The network is a crucial aspect of the whole VAHCOLI concept as decribed in (Lübken & Höffner, 2021). Most importantly crucial dynamical processes in the middle atmosphere, such as gravity waves and stratified turbulence, can be covered by VAHCOLI with sufficient temporal, vertical, and horizontal sampling and resolution. The four-dimensional capabilities provided by a VAHCOLI network allow for the performance of time-dependent analysis of the flow field, for example by employing Helmholtz decomposition, and for carrying out statistical tests regarding, for example, intermittency and helicity. Assimilation of the measured wind and temperature profiles into forecast models are a side product of this scientific goal.

We added in line 278:"Even though IAPs goal for this network is to observe crucial dynamical processes in the middle atmosphere, such as gravity waves and stratified turbulence with sufficient temporal, vertical, and horizontal sampling and resolution, it is also exceptionally well-suited to complement forthcoming Aeolus successor missions (Lübken & Höffner, 2021)."